# Phosphorylation of CENP-A on serine 7 does not control centromere function

Viviana Barra[1,6], Glennis A. Logsdon[2,3,7], Andrea Scelfo[1], Sebastian Hoffmann[1], Solène Hervé[1], Aaron Aslanian[4], Yael Nechemia-Arbely[5,8], Don W. Cleveland [5], Ben E. Black[2,3] & Daniele Fachinetti [1]

CENP-A is the histone H3 variant necessary to specify the location of all eukaryotic centromeres via its CENP-A targeting domain and either one of its terminal regions. In humans, several post-translational modifications occur on CENP-A, but their role in centromere function remains controversial. One of these modifications of CENP-A, phosphorylation on serine 7, has been proposed to control centromere assembly and function. Here, using gene targeting at both endogenous CENP-A alleles and gene replacement in human cells, we demonstrate that a CENP-A variant that cannot be phosphorylated at serine 7 maintains correct CENP-C recruitment, faithful chromosome segregation and long-term cell viability. Thus, we conclude that phosphorylation of CENP-A on serine 7 is dispensable to maintain correct centromere dynamics and function.

[1] Institut Curie, PSL Research University, CNRS, UMR 144, 26 rue d'Ulm, 75005 Paris, France. [2] Department of Biochemistry and Biophysics, Perelman School of Medicine, University of Pennsylvania, Philadelphia, PA 19104, USA. [3] Graduate Program in Biochemistry and Molecular Biophysics, Perelman School of Medicine, University of Pennsylvania, Philadelphia, PA 19104, USA. [4] Department of Chemical Physiology, The Scripps Research Institute, La Jolla, CA 92037, USA. [5] Ludwig Institute for Cancer Research and Department of Cellular and Molecular Medicine, University of California at San Diego, La Jolla, CA 92093, USA. [6] Present address: Department of Genetic Stability and Oncogenesis, Institut Gustave Roussy, CNRS UMR8200, 94805 Villejuif, France. [7] Present address: Department of Genome Sciences, University of Washington School of Medicine, Seattle, WA 98195, USA. [8] Present address: Department of Pharmacology and Chemical Biology, University of Pittsburgh Cancer Institute, School of Medicine, University of Pittsburgh, Pittsburgh, PA 15213, USA. Correspondence and requests for materials should be addressed to D.F. (email: daniele.fachinetti@curie.fr)

The accurate segregation of chromosomes during cell division is crucial to preserve the integrity of genetic information. The centromere is a key element in this process: it is the chromosomal locus that mediates the connection of the chromosomes to the mitotic spindle microtubule fibers via the formation of the kinetochore. Despite the recurrence of high order repetitive sequences at centromeric regions, in most organisms, centromeres are epigenetically specified by the H3 histone variant CENP-A (CENtromeric Protein A)[1]. CENP-A is required to maintain centromere position and assembly via a two-step mechanism[2]. First, CENP-A deposition occurs at the beginning of G1 phase via tight regulatory mechanisms[3]. Second, during interphase, it is required for the assembly of a network of centromere components named the constitutive centromere associated network (CCAN)[4]. The CCAN is then necessary to mediate assembly of the kinetochore prior to mitosis when one component, CENP-C, plays a central role[5].

Similar to other H3 variants, CENP-A is composed of a histone-fold domain that contains regions necessary for its centromere targeting (the CENP-A centromere targeting domain; CATD[6]), an extended amino-terminal tail, and a short (6-aa) carboxy-terminal tail. Both the carboxy- and amino-terminal tails of CENP-A and the CATD are required for the assembly of key components of the CCAN[2,7–9]. The carboxy-terminal tail of CENP-A directly interacts with CENP-C[7,8,10] and is crucial for its correct maintenance at centromeres[2,9]. The amino-terminal tail of CENP-A also takes part in CENP-C mobilization but indirectly via its interaction with the DNA binding protein CENP-B[2,11] and its recruitment of CENP-T[9]. Loss of either one of CENP-A's tails does not completely prevent CENP-C binding to centromeres and cell viability[2]. Altogether, this evidence suggests that both tails of CENP-A function redundantly to ensure CENP-C recruitment to the centromere and subsequent kinetochore assembly.

Post-translational modifications (PTMs) of centromeric components have recently emerged as an important factor to control centromere assembly and regulation. Specifically, CENP-A undergoes PTMs that are proposed to control its chromosomal location, its structure and stability within the nucleosome and its function[12]. The function of several of these modifications, however, remains controversial[12–15]. Great interest has been placed on the PTMs of the CENP-A amino-terminus, since this tail is highly divergent from the ones of all other H3 variants[16]. It is enriched in arginines that do not appear to be frequently modified and lacks most of the well-characterized lysines of histone H3, known to be hotspots of conserved PTMs that regulate histone function[17]. PTMs on CENP-A's amino-terminal tail such as the α-N of glycine 1 and phosphorylation of Serine 7, 17, and 19 (hereafter named S7[18,19], S16 and S18 due to first methionine digestion[18,19]) have also been detected, all of them to some extent proposed to be important for CENP-A's functionality[19–24].

In particular, the phosphorylation (ph) of CENP-A S7 has drawn much interest due to its similarities to the well-known H3 S10ph, a hallmark of mitotic entry. CENP-A S7 is phosphorylated in prophase (after H3 S10ph), reaches its highest level in prometaphase, and then starts to decrease during anaphase[18]. CENP-A S7ph is performed initially by Aurora A and maintained by Aurora B and C through telophase[20,21,25]. The exact function of CENP-A S7ph is still under debate and results are contradictory. An initial observation from Sullivan and colleagues[20] proposed a role for S7ph in the completion of cytokinesis and Aurora B localization, supported by its localization at the midbody in telophase that resembles that of the chromosomal passenger proteins (Aurora B, Survivin, Borealin, INCENP)[26]. Transient overexpression of a

non-phosphorylatable S7 (serine mutated to alanine, S7A) CENP-A variant or of a variant that mimics phosphorylation at S7 (serine mutated to glutamine, S7E) did not affect chromosome alignment, congression or separation[20]. In contrast, Kunitoku et al.[21] found that expression of CENP-A S7A results in prometaphase delay and chromosome misalignment. Using mutational analysis, the Dimitrov team further proposed CENP-A S7ph to be essential for chromosome segregation by directly controlling CENP-C binding to centromeres[22]. Indeed, quite surprisingly, they found that ectopic expression of a CENP-A S7A variant led to CENP-C loss and, consequently, to mitotic errors and cell lethality. Recently, it was found that preventing CENP-A S7ph led to sister chromatid cohesion defects but no evidence of defective kinetochore assembly was reported[24].

Despite all these proposed functions for CENP-A S7ph, the essential nature of such modification to maintain long-term mitotic function of human centromeres was never tested. Additionally, in all these reports it was unclear if the observed effects were caused by CENP-A being expressed at different levels compared to its endogenous counterpart. Furthermore, results of these experiments might be skewed due to the partial down-regulation of endogenous CENP-A achieved by siRNA. Indeed, CENP-A is a very stable and long-lived protein[27], and therefore difficult to reduce by RNAi technology[2], and its levels are critical for maintaining centromere identity and function[28–31].

Taking advantage of recently developed powerful genome editing technologies, here we describe three different approaches to test the importance of CENP-A S7ph for centromere function. First, we conditionally, rapidly, and completely remove endogenous CENP-A using an auxin-inducible degron (AID)[32] tag following ectopic expression of CENP-A variants from a unique genomic site to assess the importance of S7ph during the first few cell cycles. Second, starting from cells with one AID-tagged CENP-A allele, we use CRISPR/Cas9 to convert the other endogenous allele of CENP-A to encode a S7 non-phosphorylatable version. After induced degradation of the AID-tagged CENP-A[13], we follow centromere function in the subsequent cell cycles. Lastly, we perform endogenous CENP-A gene inactivation to deplete endogenous CENP-A in the presence of expression of a non-phosphorylatable S7 CENP-A variant[2]. In each approach, we score for CENP-A and CENP-C localization, chromosome segregation and short- or long-term viability. Altogether, our results demonstrate that CENP-A S7ph does not play an essential role in any aspect of known centromere function.

## Results

**Short-term centromere function does not require CENP-A S7ph.** We first tested if S7ph is required for maintenance of centromere function in the short-term (in the 2nd to 14th cell cycles) by using an inducible degradation system to rapidly remove endogenous CENP-A from all centromeric regions. To do this, we used a human pseudo-diploid cancer cell line DLD-1 expressing the plant F-box protein osTIR1-9xMyc[33] that we had engineered to carry one disrupted *CENP-A* allele, and the other allele tagged with *EYFP* and *AID* (CENP-A[EA])[32]. In this background, we introduced mRFP or EYFP versions of either wild-type (WT) or mutant (S7A and S7E) CENP-A proteins, constitutively expressed from a specific genomic locus (FRT site)[34] (Fig. 1a, b). The AID system ensures rapid and complete degradation of the endogenous CENP-A, following auxin hormone indole-3-acetic acid (IAA) addition[32], leaving only the ectopically-expressed versions of CENP-A at centromeres (Fig. 1c).

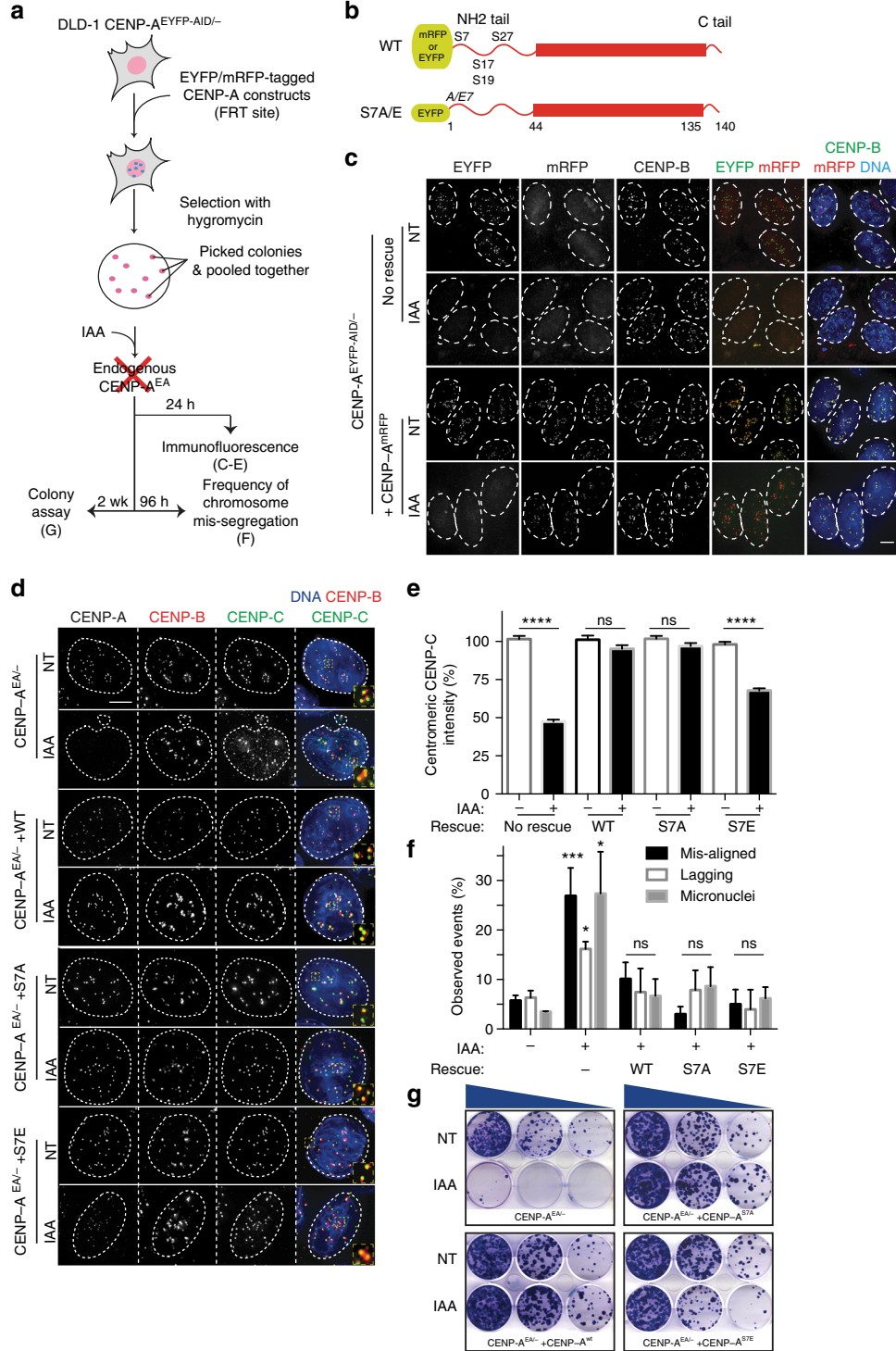

**Fig. 1** CENP-A S7ph is not required for short-term centromere function. **a** Schematics of the experiments. **b** Schematics representing the different CENP-A constructs amino-terminally tagged with EYFP or mRFP. **c** Representative immunofluorescence images showing the working principles of the system. Following IAA treatment for 24 h the EYFP signal of endogenous CENP-A disappears in both cell lines, while the mRFP signal of the exogenous CENP-A remains at the centromere (marked by CENP-B spots) in the CENP-A$^{mRFP}$ cell line. Scale bar = 5 μm. **d** Representative immunofluorescence images showing the localization of exogenous CENP-A constructs in the indicated cell lines. CENP-C staining is also shown revealing its localization in presence of exogenous CENP-A constructs (IAA treatment for 24 h). Scale bar = 5 μm. **e** Quantification of CENP-C levels of the experiments in **d**. Error bars represent the SEM of three independent experiments. Unpaired *t*-test: ****$p$ < 0.0001. **f** Quantification of mitotic errors in the indicated cell lines calculated by live-cell imaging observation after 2 days of IAA treatment. Error bars represent the SEM of four independent experiments. Unpaired *t*-test: ***$p$ = 0.0008, *$p$ = 0.012 and 0.017. **g** Representative images of crystal violet-stained colonies from the colony formation assay. Cells were grown for 14 days to test their clonogenic survival in presence or absence of endogenous CENP-A (controlled by IAA treatment). $N$ = 3. Source data for graphs shown in **e** and **f** are provided as a Source Data file

Immunoblot for CENP-A confirmed that in all the tested cell lines upon treatment with IAA the AID-tagged CENP-A was degraded, while the ectopically encoded variants were still present (Supplementary Fig. 1a). CENP-A variants that cannot be phosphorylated on S7 (S7A) or that mimic S7ph (S7E) were tested for the capability to support centromere assembly and function by measuring the centromeric level of CENP-C, levels of chromosome segregation defects, and overall cell viability. In the absence of the endogenous CENP-A following its auxin-induced degradation, CENP-A S7A and S7E correctly localized at centromeres similarly to CENP-A WT (Fig. 1d). In addition, preventing S7ph (S7A) did not affect CENP-C localization or its levels at centromeres following auxin-mediated degradation of CENP-A WT (Fig. 1d, e and Supplementary Fig. 1b). In contrast, mimicking the constitutive phosphorylation (S7E) only partially maintained CENP-C levels at centromeres (Fig. 1d, e), likely due to the low accumulation of this variant (Supplementary Fig. 1a, c). After replacing CENP-A WT with S7A, we could not detect any significant increase in mitotic errors (misaligned and lagging chromosomes observed) and micronuclei formation in the absence of endogenous CENP-A when rescued with either WT or S7A or S7E CENP-A variants, (compared to control cells) by live-cell imaging (Fig. 1f). Mitotic duration was also not impaired in S7A, and only slightly in S7E likely due to a reduced level of CENP-A/CENP-C (Supplementary Fig. 1d).

Finally, short to medium term viability was assessed by colony formation for 14 days and cell proliferation assay after continuing degradation of endogenous CENP-A (IAA treatment). CENP-A S7A mutant formed a comparable number of rescued colonies as did cells with CENP-A WT (Fig. 1g and Supplementary Fig. 1e). CENP-A S7E also supports cell viability but, in agreement with the data above, with significantly reduced percentage of colony formation. Analysis of cell proliferation also did not reveal any growth defect of CENP-A mutants (Supplementary Fig. 1f). In contrast, loss of CENP-A without any version of CENP-A rescue allele caused a significant decrease in CENP-C level, increase in mitotic errors and cell lethality, and suppression of colony survival and cell proliferation (Fig. 1d–g and Supplementary Fig. 1f), as previously described[32]. Finally, we also did not observe any increase in the midbody length measured with acetylated tubulin in CENP-A S7 mutants (Supplementary Fig. 1g), suggesting no major defects in completing cytokinesis, in contrast to an early report[20].

These results demonstrate that preventing or mimicking constitutive phosphorylation of S7 CENP-A is sufficient to rescue the total absence of endogenous CENP-A. No significant increase in mitotic errors—in terms of lagging, misaligned chromosomes and micronuclei formation—or altered CENP-C localization has indeed been detected. All these findings are inconsistent with an essential role for S7ph in overall centromere function.

Recognizing that cells that have a more unstable karyotype might be more sensitive to centromere manipulation, we next tested the influence of CENP-A S7ph in a non-diploid cell line such as HeLa, a line used in previous reports to assess the importance of such PTM[22,24]. We tagged all endogenous HeLa CENP-A alleles with an AID tag to permit its rapid, inducible degradation (Supplementary Fig. 2a, b). CENP-A WT-EYFP or S7A-EYFP constructs were then integrated by retroviral transduction and the endogenous CENP-A depleted by IAA addition. Similar to the results observed in DLD-1 cells, preventing CENP-A S7ph did not affect CENP-C binding to centromeres (Supplementary Fig. 2c) or short-term viability (Supplementary Fig. 2d–f).

We previously showed that either the amino or the carboxy tail of CENP-A is required to maintain CENP-C at centromeric regions[2,11,32]. To further investigate the possible importance of S7ph, we expressed a CENP-A/H3 chimera that contains the centromere targeting domain (CATD) and amino-terminal tail of CENP-A ($^{NH2}$H3$^{CATD}$), but lacks the CENP-A carboxy-terminal tail, known to be the site for the direct interaction of CENP-A with CENP-C[7,8,10]. This CENP-A/H3 variant has reduced levels of CENP-C and an increased rate of chromosome missegregation, although it is still capable of maintaining long-term centromere function via its amino-terminal tail[2]. We reasoned that if S7ph plays an important role in CENP-C recruitment, this importance should be enhanced in cells lacking the CENP-A carboxy-terminal tail. We integrated different variants of $^{NH2}$H3$^{CATD}$ in DLD-1 CENP-A$^{EA/-}$ cells using gene integration at the FRT site, as done above (Fig. 1a and Supplementary Fig. 3a, b). Following IAA addition, cells expressing the $^{NH2-S7A}$H3$^{CATD}$ variant that cannot be phosphorylated on S7 showed similar levels of centromeric CENP-C as did those containing the correct S7 phosphorylation site ($^{NH2}$H3$^{CATD}$) (Supplementary Fig. 3c, d). In addition, cells during 14 days without endogenous CENP-A but rescued by expression of $^{NH2-S7A}$H3$^{CATD}$ formed a comparable number of colonies to $^{NH2}$H3$^{CATD}$-expressing cells (Supplementary Fig. 3e). In contrast, CENP-A variants in which all serines known (or previously predicted) to be phosphorylated within the globular part of CENP-A N-terminal tail (S7, S16, S18 and S26) were mutated to alanines or aspartic acids ($^{NH2-S--A}$H3$^{CATD}$ and $^{NH2-S--D}$H3$^{CATD}$) only partially rescued endogenous CENP-A depletion (Supplementary Fig. 3e–g). These latter results are similar to the partial rescue of full CENP-A depletion with a H3$^{CATD}$ or with a $^{\Delta NH2}$CENP-A variant lacking the N-terminal tail[2,11]. Altogether, our results indicate that S7ph is not *per se* the cause of impairment of CENP-A N-terminus functionality and is not required to maintain the centromeric pool of CENP-C. Rather, S16 and S18 are important for CENP-A N-terminus functionality in line with the findings on the importance of S16 and S18 for centromere function[19,35].

## Endogenous CENP-A S7A does not affect centromere function.

Expression of CENP-A is known to be cell cycle regulated[36]. Thus, we next tested whether preventing CENP-A S7ph by directly modifying its endogenous locus (in order to preserve its natural promoter) has an effect on maintenance of centromere position and function. To this end, we used CRISPR/Cas9 technology in DLD-1 cells stably expressing the F-box protein TIR1-9xMyc to replace the two endogenous *CENP-A* alleles with 1) an inducible degradable CENP-A (EYFP-AID-CENP-A, hereafter referred to as CENP-A$^{EA}$) and 2) a SNAP-3xHA tagged version of either a WT CENP-A allele or a S7A CENP-A variant (Fig. 2a and Supplementary Fig. 4a). Following degradation of CENP-A$^{EA}$ via the addition of IAA, CENP-A WT- or S7A -SNAP-3xHA remained the only source of CENP-A (Supplementary Fig. 4b). Immunoblot analysis showed that the CENP-A S7A-SNAP-3xHA accumulated CENP-A to levels comparable to that of the AID-tagged protein or WT-CENP-A-SNAP-3xHA (Supplementary Fig. 4b). Further, immunofluorescence analysis revealed that CENP-A S7A expression from an endogenous locus did not impair CENP-A or CENP-C localization or their centromeric accumulation (Fig. 2b, c and Supplementary Fig. 4c), nor did it cause an increase in mitotic duration or in the rate of chromosome segregation errors (Fig. 2d, e).

Accordingly, CENP-A S7A-SNAP-3xHA fully rescued CENP-A$^{EA}$ depletion after 8 or 14 days of IAA treatment as observed by cell proliferation assay or colony formation, respectively, similarly to CENP-A WT (Fig. 2f and Supplementary Fig. 4d, e). Altogether, these results further demonstrate that preventing CENP-A S7ph does not interfere with centromere function even

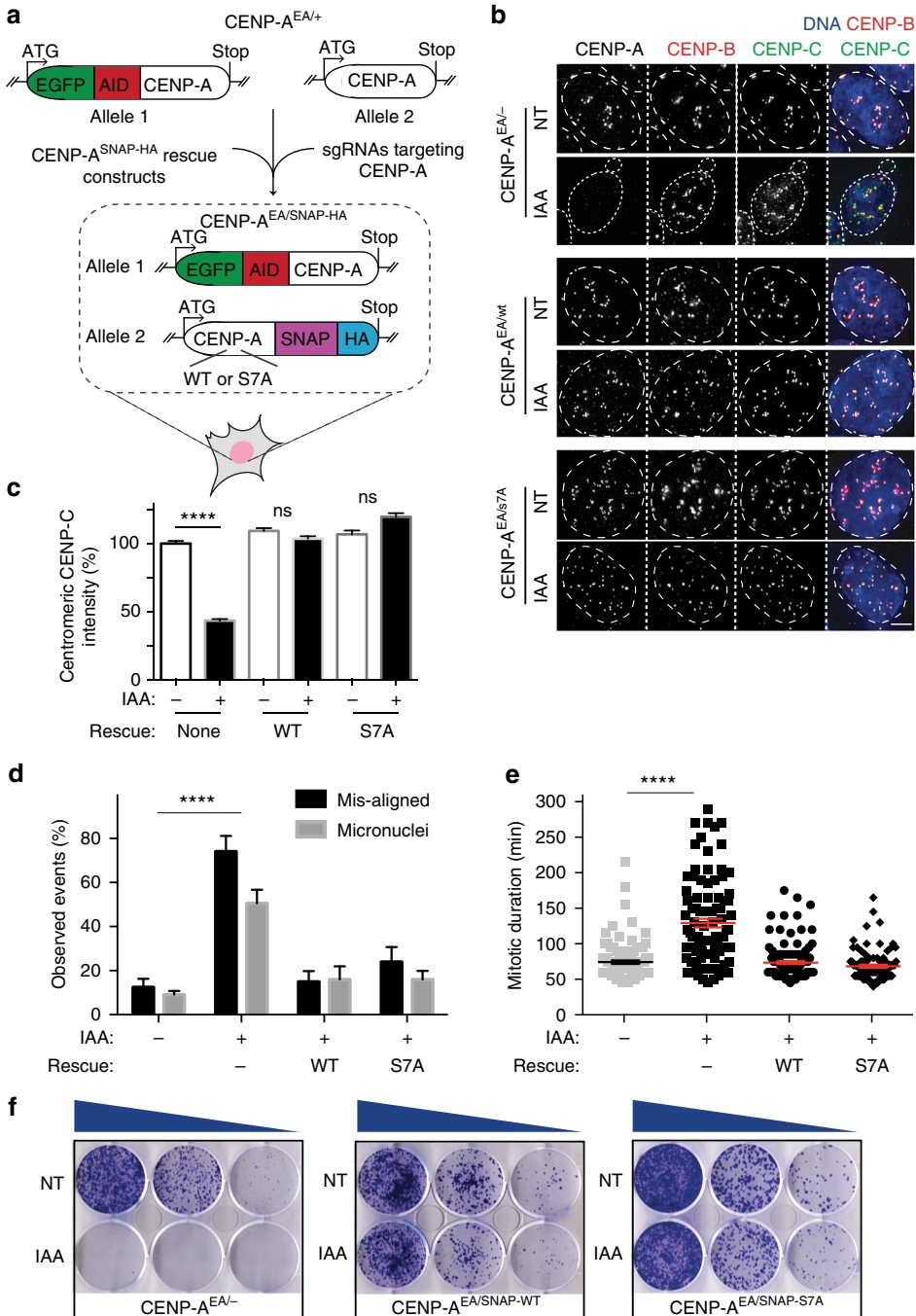

**Fig. 2** Endogenous CENP-A un-ph S7 does not affect centromere function. **a** Schematics representing the strategy used for the generation of the DLD-1 cell line. **b** Representative immunofluorescence images showing the localization of CENP-A and CENP-C in the indicated cell lines (IAA treatment for 24 h). Scale bar = 5 μm. **c** Quantification of CENP-C levels of the experiments in **b**. Error bars represent the SEM of three independent experiments. Unpaired *t*-test: ****$p < 0.0001$. **d** Quantification of mitotic errors in the indicated cell lines calculated by live-cell imaging observation after 2 days of IAA treatment. Error bars represent the SEM of three independent experiments. ANOVA test ****$p < 0.0001$. **e** Quantification of mitosis duration in the indicated cell lines. Each individual point represents a single cell. Time in mitosis was defined as the period from NEBD to chromosome decondensation. Error bars represent the SEM of three independent experiments. Unpaired *t*-test: ****$p < 0.0001$. **f** Representative images of crystal violet-stained colonies from the colony formation assay. Cells were grown for 14 days to test their clonogenic survival in presence or absence of IAA. $N = 3$. Source data for graphs shown in **c**, **d**, and **e** are provided as a Source Data file

when its expression is cell cycle regulated as for endogenous CENP-A.

**Long-term centromere function does not require CENP-A S7ph.** Finally, we tested if S7ph is required for long-term (> 100

generations) maintenance of centromere identity and function. To test this, we used RPE-1 cells engineered to have one *CENP-A* allele inactivated by gene disruption and one floxed allele (CENP-A[−/F]) that can be fully inactivated by the action of Cre recombinase[2]. CENP-A rescue constructs expressing EYFP fused to the

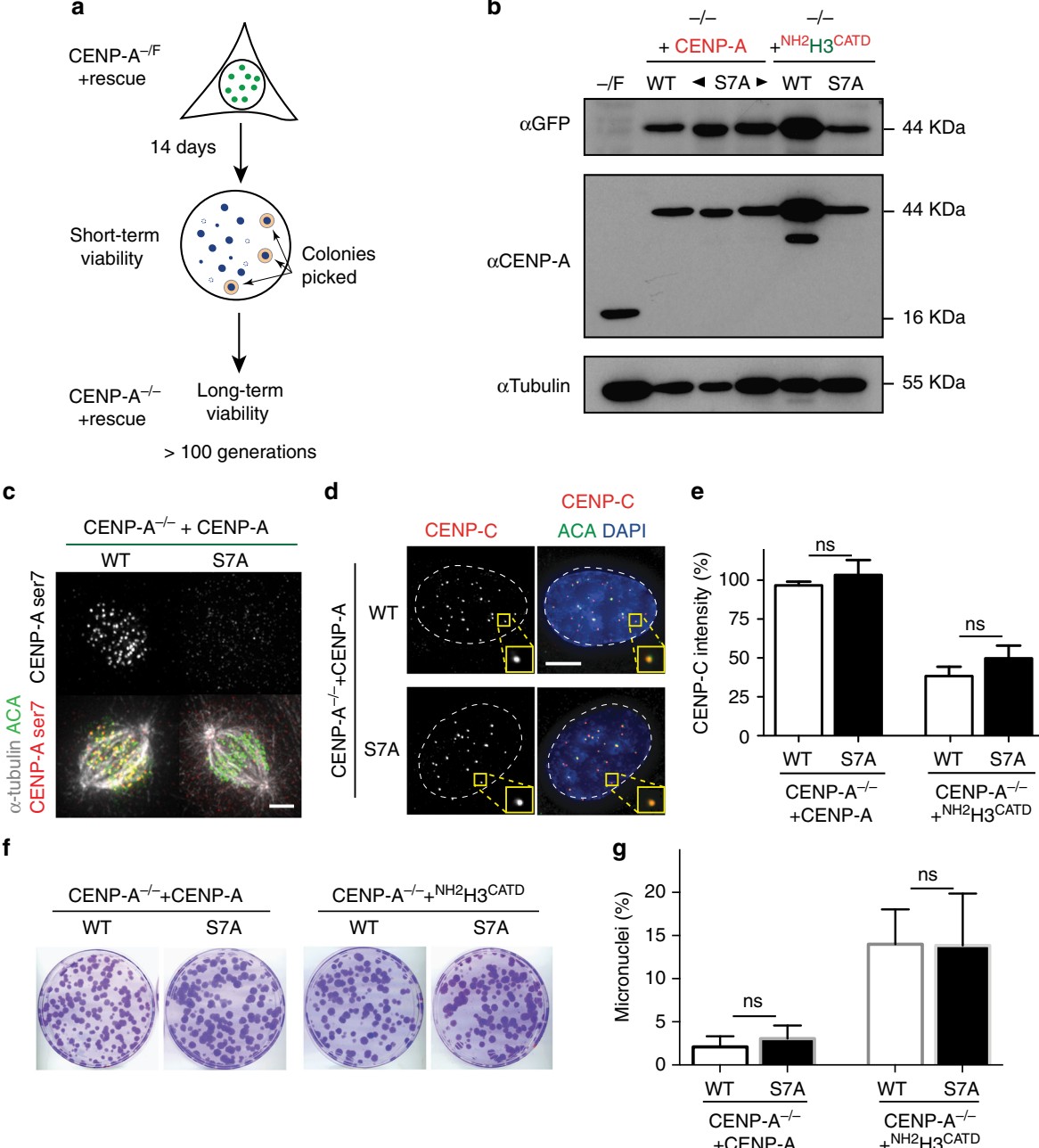

Fig. 3 CENP-A S7A completely rescues the genetic depletion of CENP-A. **a** Schematics of the experiments. **b** Immunoblots of cell extracts with antibodies against CENP-A and GFP. Antibodies against alpha-tubulin are used as a loading control. $N = 2$. **c** Representative immunofluorescence images showing that CENP-A S7A is not recognized by CENP-A Ser7ph antibody. Scale bar = 5 μm. **d** Representative immunofluorescence images showing the localization of CENP-C in the indicated cell lines. Scale bar = 5 μm. **e** Quantification of CENP-C levels of the experiments in **d**. Error bars represent the SEM of three independent experiments. Unpaired $t$-test: ns. **f** Representative images of crystal violet-stained colonies from the colony formation assay. $N = 3$. **g** Quantification of micronuclei frequency. Error bars represent the SEM of three independent experiments. Unpaired $t$-test: ns. Source data for the immunoblot shown in **b** and graphs shown in **e** and **g** are provided as a Source Data file

amino-terminus of CENP-A WT or S7A were stably expressed by retroviral integration. Additionally, we integrated rescue constructs of H3/CENP-A chimera $^{NH2}H3^{CATD}$ with WT S7 or the S7A variant to further test the importance of S7ph for CENP-C maintenance at centromeres. Following inactivation of the floxed *CENP-A* allele by Cre expression, we isolated surviving clones and tested them for long-term centromere function and viability (Fig. 3a). Immunoblot analysis confirmed the lack of endogenous CENP-A in the isolated clones (Fig. 3b). Immunofluorescence

microscopy using a commercial antibody targeting CENP-A S7ph confirmed the absence of detectable CENP-A S7 phosphorylation (Fig. 3c). Furthermore, despite all cells have the unphosphorylatable S7 CENP-A (or $^{NH2}H3^{CATD}$), CENP-C localization and its levels at centromeres were unaffected relative to cells rescued with CENP-A WT (Fig. 3d, e). Similarly, the intensities of other centromeric components including CENP-B, CENP-T and CENP-I and kinetochore proteins (Dsn1 and Hec1) were not affected (Supplementary Fig. 5a, b).

Long-term cell viability tested by colony growth assay revealed that cells expressing a CENP-A S7A mutant formed a similar number of colonies as the ones expressing a CENP-A WT rescue construct (Fig. 3f and Supplementary Fig. 5c). We also observed no differences in the duration of mitotic timing and in the rates of micronuclei formation that would indicate partial loss of centromere function, including defects in centromeric cohesion (Fig. 3g and Supplementary Fig. 5d). Accordingly, cells rescued with the unphosphorylatable S7 CENP-A variant continue to maintain a diploid state (Supplementary Fig. 5e). In addition, Aurora B and INCENP localization was unaffected in all stages of mitosis in CENP-A depleted cells rescued long-term with a CENP-A S7A variant (Supplementary Fig. 6a, b). Similarly, in DLD-1 cells Aurora B localization at the inner centromere was not affected in prometaphase-arrested cells following rapid endogenous CENP-A depletion in cells expressing either S7A or S7E (Supplementary Fig. 6c). Furthermore, we tested whether the function of Aurora B in correcting kinetochore/microtubule attachment was affected by treating cells with monastrol, a chemical inhibitor of Eg5 required for outward force generation[37]. Following monastrol wash-out, we observed that the error correction mechanism mediated by Aurora B was not impaired in RPE-1 cells rescued with a CENP-A S7A variant, while it was when cells were treated with the Aurora B inhibitor ZM447439[38] (Supplementary Fig. 6d). Only expression of the $^{NH2}H3^{CATD}$ variant, which lacks the C-terminal tail of CENP-A, produced a slight but significant decrease in Aurora B function independently of S7ph (Supplementary Fig. 6d). Altogether, our results show that the non-phosphorylation of CENP-A S7 does not affect centromere assembly (including CENP-C localization) and function, chromosome segregation fidelity, and overall cell viability even in the long-term.

## Discussion

Overall, our experimental evidence demonstrates that preventing S7ph at CENP-A does not interfere with maintenance of centromere position, assembly, or kinetochore formation via CENP-C. Nor does this particular PTM negatively affects chromosome segregation and short- (~2–14 divisions) or long-term ( >100 divisions) cell viability. Our evidence includes efforts modifying CENP-A endogenous loci (Fig. 2) or ectopically expressing a non-phosphorylatable variant in the complete absence of endogenous CENP-A (Figs. 1 and 3).

Our results (Supplementary Fig. 1) also highlight how the level of CENP-A is critical for centromere function (as in the case of the CENP-A S7E variant): too little or too much can have deleterious effects on cell viability, in agreement with previous results[2,29,30,39]. This might explain previously contradictory reports on the importance of CENP-A S7ph that were obtained with technologies that produced only partial CENP-A downregulation (achieved by RNAi) and/or transient rescue (known to lead to a range of expression levels including overexpression[40]) with CENP-A mutants[20–22,24], as observed for other PTMs of CENP-A[13–15].

What role, if any, CENP-A S7ph may provide remains an open question; however, our data strongly indicate that this posttranslational modification is not essential nor required for centromere function, in contrast to previous proposals[20–22,24]. While our data do not exclude a function for S7 CENP-A phosphorylation in some cell events such as maintenance of proper cohesion via Shugoshin[24], it argues that such function cannot be essential for long-term cell cycling and chromosome segregation. Moreover, CENP-A S7ph has never been found in mass

spectrometry data designed to identify CENP-A PTMs across the cell cycle, including during mitosis[19], suggesting that this PTM might be at low abundancy, as recently suggested[24]. We also could not detect CENP-A S7ph by mass spectrometry in any phase of the cell cycle, including mitosis, further confirming that this PTM might be present at very low levels, only occurring in a small set of nucleosomes/cells, or extremely labile during purification steps and not detectable using classical trypsin peptide digestion (Supplementary Fig. 7a–e). In contrast, our results with CENP-A/H3 chimeras ($^{NH2}H3^{CATD}$), however, have confirmed the importance of the CENP-A amino-terminal tail and its PTMs on S16/18 for centromere biology, as previously demonstrated[2,9,19,23,35]. Further studies are now necessary to address the molecular mechanisms of how CENP-A N-terminal tail phosphorylations control centromere function.

## Methods

**Cell culture conditions.** Cells were maintained at 37 °C in a 5% $CO_2$ atmosphere. DLD-1 (from S. Taylor) cells and HeLa (from H. Masumoto) cells were maintained in Dulbecco's modified essential medium (DMEM) medium containing 10% fetal bovine serum, 100 U ml$^{-1}$ penicillin, and 100 μg ml$^{-1}$ streptomycin. hTERT RPE-1 (from ATTC) cells were maintained in DMEM:F12 medium containing 10% fetal bovine serum (Clontech), 0.348% sodium bicarbonate, 100 U ml$^{-1}$ penicillin, 100 U ml$^{-1}$ streptomycin and 2 mM L-glutamine. IAA (I5148; Sigma) was used at 500 μM, Monastrol (Selleckchem, S8439) was used at 100 μM for 10 h, ZM447439 at 2 μM for 3 h, and MG132 20 μM (Calbiochem) for 3 h. All cell lines were tested for mycoplasma contamination.

**Constructs.** cDNAs used in the generation of HeLa$^{EA/-}$ TIR1 stable cell lines or hTERT RPE-1 CENP-A$^{-/F}$ were cloned into a pBabe-based vector for retrovirus generation. cDNAs used for the generation of DLD-1$^{EA/-}$ FRT TIR1 cell lines were cloned into a pcDNA5/FRT expression vector. All CENP-A and H3 mutations were generated via the Gibson assembly technique.

The plasmids used to generate the DLD-1 TIR1 cell lines via CRISPR/Cas9 [EGFP-AID-CENP-A, CENP-A(WT)-SNAP-3xHA-P2A-NeoR, and both sgRNA/Cas9 plasmids (targeting the 5′ and 3′ ends of the endogenous CENP-A gene)] were already constructed for Fachinetti and colleagues[33]. Briefly, the 5′ UTR and 3′UTR CENP-A gene regions (~800 bp each) were PCR-amplified from DLD-1 TIR1 genomic DNA. EGFP was PCR-amplified from a derivative of pBabePuro-LAP-CENP-N[4], and AID was PCR-amplified from pcDNA5-FRT-TO-H2B-AID-YFP[33]. The intronless CENP-A gene was designed using IDT's codon optimization tool, which chooses codons with a bias similar to the natural bias in the human genome, and then synthesized as a gBlock gene fragment (IDT). SNAP-3xHA-P2A-NeoR was also synthesized as a gBlock gene fragment. pUC19 was digested with EcoRI and HindIII and purified, and then, the 5′UTR, EGFP, AID, intronless CENP-A, 3′UTR, and pUC19 fragments were assembled with NEBuilder HIFI DNA Assembly Master Mix (NEB E2621). Similarly, the 5′ UTR, codon-optimized CENP-A, SNAP-3xHA-P2A-NeoR, 3′UTR, and pUC19 fragments were also assembled using NEBuilder HIFI DNA Assembly Master Mix. To generate the CENP-A(S7A)-SNAP-3xHA-P2A-NeoR repair template, QuikChange site-directed mutagenesis (Agilent) was performed on CENP-A (WT)-SNAP-3xHA-P2A-NeoR to convert S7 to an A. The sgRNA/Cas9 plasmids targeting the 5′UTR and 3′UTR of the CENP-A gene were constructed by annealing oligos and then ligating them into pX330[41] at the BbsI cut sites. For the 5′ UTR CENP-A sgRNA, the following oligos were annealed: 5′-CACCGgtgtcatgggcccgcgccgc-3′ and 5′-AAACgcggcgcgggcccatgacacC-3′. For the 3′ UTR CENP-A sgRNA, the following oligos were annealed: 5′-CACCGctgacagaaacactgggtgc-3′ and 5′-AAACgcacccagtgtttctgtcagC-3′. All plasmids were verified by sequencing using commercial primers.

**Generation of stable cell lines.** Stable DLD-1$^{EA/-}$ FRT TIR1 cell lines expressing the different CENP-A variants were generated by using the FRT/FlpIN system. Briefly, parental Flp-In TRex-DLD-1 were co-transfected with pcDNA5/FRT plasmid expressing CENP-A variants and plasmid pOG44 expressing the Flp recombinase using electroporation. Following selection in 400 μg ml$^{-1}$ hygromycin (Invitrogen), colonies were pulled together and grown for 1 week. The expression of the constructs was checked by immunoblotting. For HeLa or RPE-1 cells the CENP-A constructs were introduced by retroviral delivery. Stable integrates were selected in 5 μg ml$^{-1}$ puromycin or 10 μg ml$^{-1}$ blasticidin S and single clones were isolated and checked for construct expression by immunoblotting and immunofluorescence microscopy. Clones with similar expression levels were then pooled together. To generate DLD-1 TIR1 stable cell lines in which both alleles of CENP-A were replaced via CRISPR/Cas9 gene-editing, 400 ng of each repair template [EGFP-AID-CENP-A plasmid and CENP-A(WT or S7A)-SNAP-3xHA-P2A-NeoR plasmid] and 100 ng of each sgRNA/Cas9 plasmid (5′UTR sgRNA/Cas9 plasmid and 3′UTR sgRNA/Cas9 plasmid)[13] were co-

transfected into DLD-1 TIR1 cells[33] using Lipofectamine 2000 (Invitrogen). Seven days after transfection, 750 µg ml$^{-1}$ G418-S was added to the culture medium, and cells were cultured with G418-S for 3–4 weeks. GFP-positive cells were sorted by FACS into monoclonal lines in 96-well plates, and surviving clones were assessed by immunofluorescence microscopy and immunoblotting. Monoclonal cell lines of the indicated genotypes were verified, including for the presence of the indicated mutation, by PCR and sequencing of genomic DNA.

**Clonogenic assay.** In all experiments, 5000-1000-200 cells were plated on 6-well plates and 500 µM IAA was added the day after seeding. After 14 days, colonies were fixed 10 min in methanol and stained for 10 min using a crystal violet staining solution (1% crystal violet, 20% EtOH). The experiment was repeated at least three times for each cell line.

**Cell proliferation assay.** For WST-1 cell proliferation assay, $1 \times 10^3$ cells per well were seeded in a 96-well plate in duplicate for each day of the growth curve. IAA-treated cells were previously grown for 48 h in presence of IAA before plating. For daily measurements, cells were grown for 3 h in presence of WST-1 reagent (Sigma-Aldrich) added at 1:10 final dilution. The absorbance was measured at $\lambda = 450$ nm with a microplate reader (Fluostar) using $\lambda = 690$ nm as reference wave-length. The experiment was performed in triplicate for each cell line.

**Immunoblotting.** For immunoblot analysis, whole-cell lysates were separated by sodium dodecyl sulphate polyacrylamide gel electrophoresis, transferred onto nitrocellulose membranes (Bio-Rad), and then probed with the following anti-bodies: DM1A (α-tubulin, 1:5000, BD), CENP-A (1:1000, Cell Signalling #2186S or Abcam #ab13939), ACA (2 µg ml$^{-1}$, Antibodies-Online GmbH, 15-235-0001), GAPDH (1:2000, Cell Signalling, 14C10). Not cropped immunoblots are shown in Source Data file. The experiment was repeated twice for each cell line.

**Immunofluorescence and live-cell microscopy.** Cells were pre-extracted with PBS in 0.1% Triton-X for 1 min, fixed in 4% formaldehyde in PBS at room temperature for 10 min, then washed in 0.1% Triton-X in PBS. Cells were blocked in 2% FBS, 2% BSA, and 0.1% Tween in PBS (blocking buffer) for at least 30 min. Incubations with primary antibodies were conducted in blocking buffer for 1 h at room temperature using the following antibodies: CENP-A (3–19) mouse mono-clonal antibody (1:1000, ADI-KAM-CC006-E; Enzo), CENP-C (1:1000; Clin-isciences, PD030), CENP-B (1:1000; Abcam, ab25734), Ph-CENP-A-ser7 (Covance, 1:1000[18]), ACA (1:500; Antibodies-Online GmbH, 15-235-0001), Aur-ora B (Abcam ab2254, 1:1000), INCENP (Covance, 1:100) and HA (A-M-M#07, I. Curie platform), Acetylated tubulin (1:100; Sigma-Aldrich, T7451).

For chromosome spreads, cells were incubated with colcemid (100 ng ml$^{-1}$) for 3 h and then subjected to a hypotonic buffer treatment (with 40% of water) for 5 min at room temperature (RT) on a glass coverslip. Subsequently, cells were centrifuged for 3 min at 800 x g on a coverslip, fixed in 4% formaldehyde and then subjected to immunofluorescence as done above.

Immunofluorescence images were collected using a Deltavision Core system (Applied Precision). FITC-, Cy3-, and Cy5-conjugated secondary antibodies (Jackson lab) were used at a 1:500 dilution and incubated for 50 min. Cells were DAPI-stained and mounted with ProLong Antifade Mountant (Thermo Fisher Scientific) onto glass slides prior to imaging. For the live-cell imaging, cells were plated on high optical quality plastic slides (ibidi) treated 1 h before filming with siRDNA (1:1000) (in cells that do not stably expressing H2BmRFP) and imaged using a Deltavision Core system (Applied Precision) or an inverted spinning disk confocal (Gattaca/Nikon).

**Images quantification.** All experiments in which we performed quantifications were repeated independently at least three times. All replicates show a similar experimental variation.

Interphase centromere quantifications: Quantification of centromere signal intensity on interphase cells was done manually[2] and using two automated systems[42]. Briefly, for the manual quantifications a $15 \times 15$ pixel circle was drawn around a centromere (marked by CENP-B staining) and a circle of the same size drawn in the vicinity of the centromere (background). The integrated signal intensity of each individual centromere was calculated by subtracting the background from the intensity of the adjacent centromere. For the automated quantification, a mask of the nuclei was obtained by thresholding the DAPI channel and individual nuclei were detected using the Analyze Particles function. For each nucleus, the XY positions of centromeres were detected using the Find Maxima function on the centromere channel (CENP-B staining). Then, the mean fluorescence intensity of the channel of interest was measured in a circle of 0.3 µm radius around each centromere position. Background correction was performed by subtracting the lowest pixel intensity value within the circle to the mean intensity value of the circle. The centromeric values in a nucleus were averaged to provide the average fluorescence intensity for each individual cell and more than 30 cells were quantified per experiment.

Metaphase spreads centromere quantification: In order to measure the fluorescence intensity of Aurora B at metaphase-arrested centromeres, points were manually chosen outside the two centromeres of each individualized chromosome. A line was then drawn between the two points and the fluorescence along the line was measured for each channel using the Plot profile function in Fiji. Background correction was performed by subtracting the lowest pixel intensity value to the intensity values along the line. The line length was normalized in order to compare and pool data coming from different chromosomes; the fluorescence intensity values were extrapolated accordingly.

Midbody length measurement: The length of the acetylated tubulin bridges in dividing cells was measured with the segmented line tool in Fiji. Beta-catenin was used to discriminate cell contours.

**Immunoprecipitation and mass spectrometry.** TAP-tagged complexes were purified from chromatin and soluble fraction as previously described[4,43] by tandem affinity purification using Rabbit IgG coupled to Ultralink resin, TEV cleavage and S-protein agarose beads. Mass spectrometry was performed following peptides trypsin digestion directly on beads. For that, samples were first denatured in 8 M urea and then reduced and alkylated with 10 mM Tris(2-carboxyethyl) phosphine hydrochloride [Roche Applied Science] and 55 mM iodoacetamide [Sigma-Aldrich] respectively. Samples were then diluted using 100 mM Tris pH 8.5 to a final concentration of 2 M urea and digested with trypsin [Promega] overnight at 37 ºC.

For the phosphopeptide enrichment experiment, digested peptides were further captured by magnetic titanium dioxide beads [Pierce] according to the manufacturer's specifications. Briefly, acetonitrile was added to each sample at a final concentration of 80% prior to the addition of the titanium dioxide magnetic beads using a ratio of one microliter stock beads to ten micrograms digested peptides. The magnetic beads were incubated four times in binding buffer and four times in wash buffer prior to elution of the phosphopeptides. The eluate was lyophilized and re-suspended in 5% acetonitrile/0.1% formic acid (buffer A).

Digested peptides were pressure-loaded onto split column placed in line with a 1200 quaternary high pressure liquid chromatography (HPLC) pump [Agilent Technologies] and the eluted peptides were electrosprayed directly into an LTQ Orbitrap Velos mass spectrometer [Thermo Scientific]. A 180 min elution gradient was run twice. The MS/MS cycle consisted of one full scan mass spectrum (400–1600 mz$^{-1}$) at 60 K resolution followed by ten data-dependent collision induced dissocation (CID) MS/MS spectra. Charge state exclusion was enabled with + 1 and unassigned charge states rejected for fragmentation. Application of mass spectrometer scan functions and HPLC solvent gradients were controlled by the Xcalibur data system [Thermo Scientific].

MS/MS spectra were extracted using RawXtract (version 1.9.9). MS/MS spectra were searched with the ProLuCID algorithm[44] against a human UniProt protein database downloaded on 25-03-2014 that had been supplemented with common contaminants and concatenated to a decoy database in which the sequence for each entry in the original database was reversed. The ProLuCID search was performed using full enzyme specificity (cleavage C-terminal to Arg or Lys residue), static modification of cysteine due to carboxyamidomethylation (57.02146) and variable modification of serine/threonine/tyrosine due to phosphorylation (79.9663). The data were searched using a precursor mass tolerance of 50 ppm and a fragment ion mass tolerance of 600 ppm. The ProLuCID search results were assembled and filtered using the DTASelect (version 2.0) algorithm[45]. DTASelect assesses the validity of peptide-spectra matches using the cross-correlation score (XCorr) and normalized difference in cross-correlation scores (deltaCN). The search results are grouped by charge state and tryptic status and each sub-group is analyzed by discriminant analysis based on a non-parametric fit of the distribution of forward and reversed matches. Only modified peptides were considered and a minimum of one peptide was required for each protein identification. All peptide-spectra matches had <10 ppm mass error. The peptide false-positive rate was below one percent for all experiments. Phosphosite localization was performed using AScore. AScore is a modification localization score. A value > 13 indicates 95% confidence and a value of > 20 indicates 99% confidence[46].

**Reporting summary.** Further information on experimental design is available in the Nature Research Reporting Summary linked to this article.

## Data availability

All data supporting the findings of this study are available within the article and its Supplementary Information files or from the corresponding author upon reason-able request. The computer codes for image quantifications are available from the corresponding author upon request. The source data underlying Figs. 1e, f, 2c–e, 3b, e, g and Supplementary Figs. 1a, 2a, 3b, 4a, and 7 are provided as a Source Data file. The raw data of the mass spectrometry (MS) results are available from the corresponding author upon request. The MS proteomics data of the phospho-peptide enrichment have been deposited to the ProteomeXchange Consortium via the PRIDE partner repository under the dataset identifier PXD012163.

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

## Acknowledgements

We would like to thank A. Echard (I. Pasteur, Paris), C. Bartocci (Institut Curie, Paris), the Fachinetti, Basto and Drinnenberg team members for helpful suggestions and technical help. We would also like to thank E. Selzer (UPenn) for his work in generating cell lines and A. Desai (Ludwig, La Jolla) and F. Perez (Institut Curie, Paris) for providing reagents. We also thank J. R. Yates III (SCRIPPS, CA) for the usage of the MS machine, the FACS facility in the Sanford Consortium for Regenerative Medicine (La Jolla, CA), the microscopy platform at Institut Curie supported by the French National Research Agency through the «Investments for the Future » program (France-BioImaging, ANR-10-INSB-04), the Fondation pour la Recherche Médicale (FRM No. DGE20111123020) and the Canceropôle-IdF (No. 2012-2-EML-04-IC-1 & No. 2011-1-LABEL-IC-4). This work was also supported by NIH R01-GM082989 (B.E.B.). G.A.L. also acknowledges the UPenn Cell and Molecular Biology Training Grant (NIH T32-GM007229). D.F. receives salary support from the CNRS. D.F. has received support by Labex « CelTisPhyBio », the Institut Curie and the ATIP-Avenir 2015 program. This work has also received support under the program «Investissements d'Avenir» launched by the French Government and implemented by ANR with the references ANR-10-LABX-0038 and ANR-10-IDEX-0001-02 PSL.

## Author contributions

G.A.L. performed genome editing of CENP-A in Fig. 2. Y.N.A. and A.A. performed mass spectrometry experiments. D.F., V.B., A.S., S.H., S.H. performed all the remaining experiments. D.F. and V.B. analyzed the data. D.F. and V.B. wrote the manuscript and all authors contributed to manuscript editing. B.E.B. and D.W.C. provided important suggestions on the study design. D.F. conceived the experimental design.

**Additional information**

**Competing interests:** The authors declare no competing interests.

