## [Peer Review File · Nature Communications]

Reviewer #1 (Remarks to the Author):

The manuscript from Barra et al explores the importance of phosphorylation on the serine 7 residue of the histone H3 variant, CENP-A. CENP-A is crucial in the process of cell division because it epigenetically identifies centromeres, the chromosomal locus where kinetochore proteins bind to mitotic spindle fibers. In particular, the amino-terminal tail of CENP-A is important for the recognition and recruitment of the CCAN proteins including CENP-C, CENP-B and CENP-T. Several conflicting pieces of data exist for the function of CENP-A S7 phosphorylation. In this paper, the authors aim to define the role of p-CENP-AS7 in centromere function using CENP-A knockout and replacement experiments, which is a much more powerful and decisive approach than has been used previously. Using DLD-1 and HeLa cells engineered to degrade endogenous CENP-A, the authors demonstrate that the introduction of WT or mutant (S7A and S7E) did not differentially affect CENP-A localization, CENP-C localization, mitotic errors or overall cell viability. The unphosphorylated (S7A) or constitutively phosphorylated (S7E) mutants are sufficient to rescue the absence of endogenous CENP-A. Next, the authors replaced the two endogenous CENP-A alleles in DLD-1 cells with an auxin inducible degradable CENP-A and a SNAP-3xHA tagged version of WT or S7A CENP-A. The results reveal that CENP-A S7A expressed from the endogenous locus did not disrupt CENP-C localization or accumulation. Finally, the authors used RPE-1 cells with depleted endogenous CENP-A and rescued the cells with EYFP fused CENP-A WT and S7A constructs. The results showed that long-term cell viability is unaffected by inability of CENP-A to be phosphorylated at S7. Overall, the data show that phosphorylation of CENP-A at the serine 7 residue is non-essential to centromere assembly and function. And so, despite almost 15 years of study, CENP-A S7 phosphorylation has no effect on centromere function. The gene replacement methods used in this paper, and the use of multiple cell lines makes the current study very compelling compared with previous work. There are some minor issues that need to be addressed, but overall the experiments are very well done. It is a little disappointing that the authors did not present any positive data in the manuscript, and this brings up the issue of whether presenting negative results are sufficient for publication. I do think it is important to communicate these results, given the long history of the study of S7 phosphorylation of CENP-A.

Major Points:

1. The original phenotype described for S7 was midbody defects (Zeitlin et al 2001). The authors should check this as well.
2. Kunitoku also showed that Aurora B localization to the inner centromere was reduced, and the duration of mitosis was extended in S7 mutants. The authors should also test these phenotypes.
3. The authors show no chromosome missegregation defects in figure 1F, using DLD-1 cells rescued by CENP-A driven off an exogenous promoter (FRT integrated). Do the DLD-1 cell lines with the CENP-A-SNAP S7A mutant show a similar lack of chromosome segregation defects? This is a superior setup because the S7A-CENP-A is expressed from the endogenous promoter.

4. The colony formation assays provide information about overall viability, but do not provide quantitative assessment of cell proliferation. The authors should conduct cell quantitative proliferation assays.
5. In Fig S2C, endogenous CENP-A intensity is shown only in none and S7E mutant; instead, it would be more relevant if authors show the levels of wt and mutant levels.

Minor issues

6. Statistics are required in Figure 6A and 6B.
7. Quantitation of the colony formation assays is needed across biological replicates in all figures.
8. Authors used S->E as phospho mimetic mutants in the first experiments but in subsequent experiments, they switched to S->D mutant. This requires an explanation.

Reviewer #2 (Remarks to the Author):

Barra et al present data on the physiological function of a post-translational modification (PTM) thought to be required for centromere establishment and action. The importance of the specific modification, phosphorylation of serine 7 on CENP-A, has been the subject of some debate in the literature. The data presented in this manuscript purports to conclusively show that the modification is of no specific consequence. While a negative result, ruling out a definitive function for the PTM would bring considerable clarity to the field, and allow one to better define the PTMs that are genuinely required for centromere function.

Using a combination of gene-replacement and depletion-rescue experiments, the authors convincingly show that an S7 non-phosphorylatable version of CENP-A effectively mimics the wild-type protein in all tested approaches. The data and associated statistical analyses presented are convincing, and I do not think any additional experiments are required. Less can be made of the results pertaining to the S7E mutant as there is now ample evidence that such phospho-mimetic mutations are of widely varying efficacy; this does not however detract from the main conclusion of the paper.

My only concern is with the mass-spec analyses of phosphopeptides presented in supplemental figure 1. It is not clear if the failure to identify S7 modifications is because they do not exist, or that the appropriate peptide is not detected by the instrument. It would be helpful to know if the unmodified peptide containing S7 can be detected in an unbiased (i.e. pre-phospho enrichment step). If it cannot, the conclusions drawn from this experiment will need revision.

Point by point response to reviewer comments on Barra et al., submitted to Nature Communications

(Our answers are below in red and italic; reviewer's comments are in black).

Reviewer #1 :

Overall, the data show that phosphorylation of CENP-A at the serine 7 residue is non-essential to centromere assembly and function. And so, despite almost 15 years of study, CENP-A S7 phosphorylation has no effect on centromere function. The gene replacement methods used in this paper, and the use of multiple cell lines makes the current study very compelling compared with previous work. There are some minor issues that need to be addressed, but overall the experiments are very well done.

We thank this Reviewer for the positive and favourable comments on our work. We have indeed tried to take advantage of more recent and efficient techniques to solve once and for all the impact of CENP-A S7 phosphorylation in cell division.

It is a little disappointing that the authors did not present any positive data in the manuscript, and this brings up the issue of whether presenting negative results are sufficient for publication. I do think it is important to communicate these results, given the long history of the study of S7 phosphorylation of CENP-A.

We do agree with this Reviewer that presenting negative results is generally uncommon for publication, but the role of CENP-A phosphorylation on S7 has puzzled the researchers of the field for several years and it still remains controversial. For this reason, as the reviewer agrees, we think that our data are important to be communicated to finally and undoubtedly rule out the essential nature of such modification in centromere biology.

Major Points:

1. The original phenotype described for S7 was midbody defects (Zeitlin et al 2001). The authors should check this as well.

We thank this Reviewer for the suggestion. We performed the experiment and we observed that neither the impairment nor the constitutive phosphorylation of CENP-A S7 increased the length of the midbody measured by the length of the acetylated tubulin track (new sup figure 1g). Nevertheless, there is a mild but significant reduction in the CENP-A S7E expressing cells. However, the experiment that we performed here does not want to be a definitive measurement of defects in cytokinesis (that is outside of the scope of this manuscript), but it gives us an indication that formation and resolution of the midbody is not strongly (or at all) controlled by CENP-A S7 ph, conversely to what was shown by Zeitlin et al. J.Cell.Biol. 2001. This is a good example of how an improved technology could bypass previous technical limitations as the one used in the original study from Kevin Sullivan's group.

2. Kunitoku also showed that Aurora B localization to the inner centromere was reduced, and the duration of mitosis was extended in S7 mutants. The authors should also test these phenotypes.

As suggested by this Reviewer, we monitored the duration of mitosis in CENP-A depleted cells rescued with CENP-A S7A (ectopically or mutated at its endogenous locus) or S7E versus WT cells and we did not notice any significant increase in mitotic timing, except for a slight (but significant) increase in the S7E mutant (new figure 1d). However, considering that S7E cells express low levels of CENP-A (Sup Fig. 1c) and CENP-C (Fig. 1e) this was expected. Nevertheless, none of these mutants has a comparable level of increase in mitotic timing to the one in CENP-A null cells, as one might expect for an “essential modification”. These results confirmed our previous finding in original sup fig. 6C (now 5d) and new figure 2e that CENP-A S7 phosphorylation does not negatively impact mitosis duration.

In addition, we stained and measured Aurora B localization at the inner centromeres on prometaphase-arrested cells (new sup figure 6c). Also in this case, modifications of CENP-A S7 ph do not compromise Aurora B localization, supporting our data following long-term CENP-A depletion and rescue (original sup fig. 6E-G, now sup figure 6a, b and d)

3. The authors show no chromosome missegregation defects in figure 1F, using DLD-1 cells rescued by CENP-A driven off an exogenous promoter (FRT integrated). Do the DLD-1 cell lines with the CENP-A-SNAP S7A mutant show a similar lack of chromosome segregation defects? This is a superior setup because the S7A-CENP-A is expressed from the endogenous promoter.

We thank this Reviewer for this suggestion. We performed three independent experiments of live cell imaging for DLD-1 cells harbouring CENP-A-SNAP S7A mutant expressed from the endogenous promoter and once again we do not observe an increase in chromosome segregation defects (new figure 2d) or an increase in mitotic timing (new figure 2e).

4. The colony formation assays provide information about overall viability, but do not provide quantitative assessment of cell proliferation. The authors should conduct cell quantitative proliferation assays.

As this Reviewer suggested, we checked quantitatively the proliferation of all CENP-A S7 mutants (used in the original colony assays) under IAA condition and we did not observe any significant difference in comparison with control cells expressing endogenous CENP-A (NT). However, we could see a significant difference in cell proliferation in cells lacking CENP-A without any rescue. These results confirmed that CENP-A S7 phosphorylation is not necessary for CENP-A cell viability (new sup figures 1f, 2f, 3g and 4d).

5. In Fig S2C, endogenous CENP-A intensity is shown only in none and S7E mutant; instead, it would be more relevant if authors show the levels of wt and mutant levels.

As this Reviewer suggested we now show the quantification of endogenous CENP-A intensity in all the mutants and wt cells (new sup figure 1c). Although we observed a bit of variability in the amount of CENP-A between our experiments, only S7E shows a significant decrease of CENP-A level that could therefore explain the reduction of CENP-C observed in figure 1e

Minor issues

6. Statistics are required in Figure 6A and 6B.

We apologize for the oversight. The statistics have been added to the sup figures 5a, b. The statistical analysis confirmed that we could not observe any difference between WT and S7A CENP-

A variant in the intensity of centromere and kinetochore proteins, but only between cells carrying the full-length CENP-A versus the CENP-A mutant containing only N-terminal tail (first 29 amino acids) and the CATD domain (^{NH2}H3^{CATD}).

7. Quantitation of the colony formation assays is needed across biological replicates in all figures.

We again apologize for the oversight. Quantifications of the colony formation assays (of at least 3 independent experiments) have now been added for all the cell lines tested and the respective results are now shown in sup figures 1e, 2e, 3f, 4c and 5c.

8. Authors used S->E as phospho mimetic mutants in the first experiments but in subsequent experiments, they switched to S->D mutant. This requires an explanation.

We apologize to this Reviewer for the confusion. We used two types of phospho mimetic mutants: for the CENP-A full length we have only used the mutant S → E (figure 1), but for the ^{NH2}H3^{CATD} (sup figure 3) we have converted all 4 Serines to Aspartic Acid (D).

Reviewer #2

Barra et al present data on the physiological function of a post-translational modification (PTM) thought to be required for centromere establishment and action. The importance of the specific modification, phosphorylation of serine 7 on CENP-A, has been the subject of some debate in the literature. The data presented in this manuscript purports to conclusively show that the modification is of no specific consequence. While a negative result, ruling out a definitive function for the PTM would bring considerable clarity to the field, and allow one to better define the PTMs that are genuinely required for centromere function.

We thank this Reviewer for the positive comments on our data. We do agree with this Reviewer that despite these are negative results, however they are important for the scientific community studying centromere biology.

Using a combination of gene-replacement and depletion-rescue experiments, the authors convincingly show that an S7 non-phosphorylatable version of CENP-A effectively mimics the wild-type protein in all tested approaches. The data and associated statistical analyses presented are convincing, and I do not think any additional experiments are required. Less can be made of the results pertaining to the S7E mutant as there is now ample evidence that such phospho-mimetic mutations are of widely varying efficacy; this does not however detract from the main conclusion of the paper.

We thank this Reviewer for the positive comment.

My only concern is with the mass-spec analyses of phosphopeptides presented in supplemental figure 1. It is not clear if the failure to identify S7 modifications is because they do not exist, or that the appropriate peptide is not detected by the instrument. It would be helpful to know if the unmodified peptide containing S7 can be detected in an unbiased (i.e. pre-phospho enrichment step). If it cannot, the conclusions drawn from this experiment will need revision.

This reviewer raises a good point. The appropriate peptide that contains phospho S7 could not be detected by the instrument, but this was true also when we looked at the unmodified peptide containing S7 in an unbiased approach. A likely explanation is that the peptide is very basic and perhaps trypsin is not the right enzyme to use in this case. It is important to note that the detection

of the unmodified and modified peptides are independent events, meaning that the presence or absence of one says nothing about the other, despite the non-phosphorylated peptides tend to be easier to detect. Nevertheless, we agree with the reviewer that our conclusion based on this experiment should be toned down. For clarity, we have now decided to include in our original data the analysis done without the enrichment of phospho peptide and to move the whole figure (original sup fig 1) to the discussion (new sup figure 7) and to add the following sentence: “We also could not detect CENP-A S7ph by mass spectrometry in any phase of the cell cycle, including mitosis, suggesting that this PTM might be present at very low levels, only occurring in a small set of nucleosomes/cells, or extremely labile during purification steps and not detectable using classical trypsin peptide digestion (sup figure 7)”.

However, our overall message is not dependent on the MS data, thus diminishing the importance of the conclusions that were drawn from it will not weaken the overall story.

Reviewer #1 (Remarks to the Author):

The authors have done an excellent job of addressing this reviewer's concerns with the previous version of the manuscript. I have no further comments.

Reviewer #2 (Remarks to the Author):

I'm happy with the authors' responses to my original comments, and fully support publication of this manuscript as is.

Barra et al., Responses to the Reviewers

(in red and italics are the Authors' responses)

Reviewer #1 (Remarks to the Author):

The authors have done an excellent job of addressing this reviewer's concerns with the previous version of the manuscript. I have no further comments.

We thank this Reviewer for the positive comment. We think that the manuscript has greatly benefitted from this Reviewer's suggestions.

Reviewer #2 (Remarks to the Author):

I'm happy with the authors' responses to my original comments, and fully support publication of this manuscript as is.

We are grateful to this Reviewer for the favourable comments and for the helpful suggestions given about the previous version of the manuscript.